# Prostate Cancer Screening with PSA: Ten Years’ Experience of Population Based Early Prostate Cancer Detection Programme in Lithuania

**DOI:** 10.3390/jcm9123826

**Published:** 2020-11-26

**Authors:** Ausvydas Patasius, Agne Krilaviciute, Giedre Smailyte

**Affiliations:** 1Laboratory of Cancer Epidemiology, National Cancer Institute, LT 08660 Vilnius, Lithuania; giedre.smailyte@nvi.lt; 2Department of Public Health, Institute of Health Sciences, Faculty of Medicine, Vilnius University, LT-03101 Vilnius, Lithuania; 3Division of Clinical Epidemiology and Aging Research, German Cancer Research Center (DKFZ), 69120 Heidelberg, Germany; a.krilaviciute@dkfz-heidelberg.de

**Keywords:** prostate cancer, screening, PSA, population-based

## Abstract

The aim of this study is to report key performance estimates from the ten years of a population-based prostate cancer screening programme in Lithuania. Retrospective analysis of screening activities recorded in 2006–2015 among men aged 50–74 years was performed. We estimated screening coverage, cancer detection rate, compliance to biopsy, and positive predictive values in each screening round inside and outside the target population. In the first 10 years of screening, 16,061 prostate cancer cases were registered within the screening programme, 10,202 were observed among screened men but reported outside the screening programme, and 1455 prostate cancers were observed in a screening-naïve population. Screening cover reached up to 45.5% of the target population in the recent rounds. The proportion of prostate specific antigen (PSA) test-positive men decreased from 16.9% in 2006 to 10.7% in 2014–2015. Up to 40.0% of PSA test-positive men received a biopsy, of whom 42.0% were positive for prostate cancer. The cancer detection rate was 10.4−15.0% among PSA test-positives and 1.4–1.9% among screened individuals. Screening participants were more likely to be diagnosed with organ-confined disease as compared to non-participants. Despite the unorganized screening practices being employed and low coverage per screening round, 70% of the target population were screened at least once in the first 10 years of screening.

## 1. Introduction

Prostate cancer is the third most common cancer in men [1] and the widespread implementation of prostate specific antigen (PSA) testing has changed the epidemiologic situation of prostate cancer worldwide [2]. Leading to overdiagnosis and overtreatment, PSA-based screening has reduced prostate cancer mortality. Screening practices using PSA are ongoing in many countries, however the benefits of such screening remain debated across communities [3]. In 2011. the United States Preventive Service Task Force (USPSTF) recommended against prostate cancer screening in all age groups [4]. The United States discontinued prostate cancer screening in the same year, but this recommendation was updated in 2018, when it was recommended to screen 55–59-year old men based on a shared decision between the patient and physician [5]. Recently, the European Urology Association expressed their interest for PSA-based prostate cancer screening at the population level in European countries [6]. However, appropriate screening measures to avoid overdiagnosis and overtreatment are yet to be determined.

Population-based screening for prostate cancer is already ongoing in one of a few relatively small Northern-Europe countries. Lithuania, a country with 2.8 million inhabitants in 2019 launched nation-wide PSA-based Early Prostate Cancer Detection Programme (EPCDP) in 2006 that works on an invitation by opportunity basis. According to the programme description, the objective of this programme is to reduce prostate cancer mortality in the target age male population and to diagnose prostate cancer at earlier stages of disease. Since then, Lithuania has experienced a tremendous increase in prostate cancer incidence. Age-standardized prostate cancer incidence peaked to 279 per 100,000 men (European Standard) in 2007 and remained above 160 cases per 100,000 until 2016, the year of data extraction for recently published analysis [7]. During the first years of the EPCDP screening programme, prostate cancer incidence exceeded that observed in the United States in the 1990s [8], making them the highest prostate cancer incidence peaks ever seen in a country. In the most recent analysis, prostate cancer incidence rates in Lithuania were reported to be the highest in the world [2]. While the PSA screening programme in Lithuania has been implemented for more than 10 years, the data published so far are scarce, covering only an analysis of incidence and mortality changes [7,8].

In this paper, we aimed to report for the first time the results of the ten years from the ongoing nation-wide PSA-based screening programme in Lithuania.

## 2. Materials and Methods

### 2.1. Early Prostate Cancer Detection Programme (EPCDP)

Screening for prostate cancer in Lithuania is offered by general practitioners when visited for any reason. EPCDP is funded by the National Health Insurance Fund (NHIF) that covers health system activities for almost all residents (98%) of the country. Figure 1 shows a flow chart for EPCDP between 2006 and 2016. A PSA test was offered for all men aged 50–74 and men aged 45–49 with a family history of prostate cancer. Men with a PSA level of ≥3 ng/mL were consulted by a urologist who referred them for biopsy after suspicious findings in a digital rectal examination (DRE). Men with PSA < 3 ng/mL, as well as those who refused biopsy or those with biopsy results indicating no malignant changes, were referred to the next screening round. Screening was offered annually between 2006 and 2009 and biennially between 2010 and 2016. Screening guidelines were modified in 2017 where changes in the target population age and implementation of age-specific PSA cut-offs were introduced. In this analysis, we focus only on years 2006–2015, which corresponds to the first seven screening rounds. The year 2016 was excluded as it comprised only half of the screening round.

### 2.2. Data Sources

The NHIF database contains demographic data and entries on the primary and secondary healthcare services, emergency and hospital admissions, and prescriptions of reimbursed medications for chronic diseases [9]. We based our analysis on data extracted from the NHIF database between the 1st of January 2006 and the 31st of December 2015. As such, the target population size among those aged 50–74 years old per screening round was extracted together with exact date and PSA test results (<3 ng/mL, ≥3 ng/mL), date and results of biopsy histopathological examination, vital status at the end of follow-up, date of death, and date of emigration when applicable for all men regardless of age.

For a small proportion of men who had multiple screening PSA or biopsies per screening round, only one screening test with most advanced findings was considered. Findings from the repeated PSA tests and biopsies after already confirmed prostate cancer diagnosis were excluded from the analyses.

In addition, data from the NHIF database was linked to Lithuanian Cancer Registry database that contains information on age at cancer diagnosis, date of diagnosis and tumor stage according to the TNM classification of malignant tumors TNM classification. Linkage was based on a unique personal identification number that is used throughout all information systems in Lithuania.

We based this analysis on the first seven screening rounds conducted in the calendar years 2006, 2007, 2008, 2009, 2010–2011, 2012–2013, and 2014–2015. For each screening round, the following indicators were calculated among individuals aged 50 to 74 years: screening coverage, proportion of PSA test-positives and test-negatives, ratio between number of biopsies and PSA test-positives, ratio between detected cancer cases and PSA test-positives, and ratio between detected cancer cases and screened persons. The same indicators were calculated among men 45–49 years except from screening coverage that was not available due to unknown family history of prostate cancer. Prostate cancer diagnosis in men who participated in screening programme at least once, but cancer was not reported as diagnosed within screening programme were assigned to the screening. Prostate cancer stage distribution was calculated for cases registered in the screening, for cases not registered in the screening among those who participated at least once, and for prostate cancers in a screening-naïve population.

### 2.3. Ethical Approval

This study was performed in line with the principles of the Declaration of Helsinki. Approval was granted by Vilnius regional biomedical research ethics committee, approval number 158200-16-879-388 on 28th November 2016.

## 3. Results

Between 2006 and 2015, 655,487 men belonged to the target age group of screening (born between the years 1931 and 1966) and 459,667 (70.1%) of them were screened for prostate cancer at least once. A total of 1,179,283 PSA tests were performed and 1,044,448 tests were performed within the target population.

As seen in the flow chart of the study cohort (Figure 2), 16,061 (56.7%) prostate cancer cases were registered within the screening programme and 10,202 (38.2%) prostate cancers were observed among men who participated in the screening programme at least once, however, prostate cancer was not registered within the program. Overall, 5.8% (=26,896 out of 459,667) of screened men were diagnosed with prostate cancer as compared to 0.7% (=1455 out of 195,820) among screening-naïve individuals. In the group of registered as non-screen detected cancers, 60% of cases were PSA test-positives and prostate cancer diagnosis was reported shortly after the last screening test (median 97 days).

The main performance indicators of the prostate cancer screening programme in Lithuania in the first seven screening rounds are shown in Table 1. The participation rate varied from 22.4% to 28.8% between annual screening rounds and from 39.5% to 45.5% between biannual screening rounds. The proportion of PSA test-positive men decreased from 16.9% in first round to 10.7% in the seventh round. Between 28.4% and 39.2% of PSA test-positive men received a biopsy of whom 35.9–42.0% were positive for prostate cancer. The prostate cancer detection rate among PSA test-positive individuals varied between 10.4% and 15.0%.

Screening practices among men aged 45 to 49 years are reported in Appendix A. PSA ≥ 3 ng/mL were detected in 6.3–7.3% of tested men and prostate cancer was confirmed in 0.3–1.5% of tested individuals. Screening practices outside target groups are shown in Figure 3. A steadily decline in unnecessary screening activities was observed within screening rounds over the time. Among the screened men between 50 and 74 years 3.6% received more than one PSA test in the first round (3339 out of 92,896 = 3.6%) and 0.7% in the seventh screening round (1511 out of 223,958 = 0.7%). Between 236 (first round) and 107 (last round) men received more than one biopsy per screening round. Among older than 74-year-old men, a decline in screened individuals from 1751 to 307 was observed. As screening test PSA test was used in men with already detected prostate carcinoma. Number of younger than 45-year-old men who were screened within the programme varied between 236 and 321 in first five rounds and dropped to 12 individuals in the last round.

Figure 4 shows the stage distribution of prostate cancer in screen-detected and screened registered as non-screen-detected and unscreened individuals. Among the unscreened men, 12.4% of cases were diagnosed with stage IV disease. In contrast, 1.2% and 2.6% were stage IV among prostate cancer registered in screened screen-detected and outside the screening programme in men who participated at least once.

## 4. Discussion

We present the first results from the nation-wide PSA-based screening programme over ten years between 2006 and 2015. In these years, prostate cancer screening in Lithuania covered less than half of the target population in each screening round. Still, 70% of men of age 50–74 years have been screened at least once over the period of 10 years. The PSA test was positive in less than 17% of tested men, among whom 9–13% were diagnosed with prostate cancer. Majority of screening-detected prostate cancers were detected at early stage.

Reduction in mortality is the main indicator for screening programme effectiveness. In this analysis, we do not report data on mortality as we focused on reporting the screening activities and prostate cancer detection rate in the initial screening years. A longer follow-up period might be needed to observe the effect on mortality rates. Changes in mortality trends have been reported previously [7].

Screening uptake (coverage) is often described as most important factor determining the success of screening programme [10,11]. However, prostate cancer screening when performed in all men without an individual risk-benefit assessment may suffer from substantial overdiagnosis. Therefore, screening uptake in this analysis should be interpreted with caution. Prostate cancer screening coverage in Lithuania increased with every screening round but had not reached half of the target population, that might be due to the unorganized nature and the lack of monitoring system of screening services. An increase in coverage when screening programme switched from annual to biennial screening intervals most likely shows prolonged screening interval effect. According to Medical Expenditure Panel Survey in 2006 in United States, 49.7% of men aged 50–74 years received screening PSA test [12] that is comparable to that observed in Lithuania. In Japan, municipality-based prostate cancer screening coverage reached only 20% of the target population [13].

The number of performed PSA tests increased in conjunction with growing screening attendance over time. Notably, screening was also observed outside target age groups and screening intervals. Such screening practices decreased stepwise with every screening round. It is notable that a small proportion of screening services was performed for patients with already detected prostate adenocarcinoma that may correspond to diagnostic purposes or active surveillance for these cases.

The positivity rate for PSA testing is highly dependent on the cut-off value. It is known that serum PSA concentration level strongly increases with age [14]. Since 2017, age-specific PSA cut off levels were implemented in the screening programme to decrease the costs of screening with possible decrease the likelihood of over-detection and false-positive PSA in older ages. However, in the first seven rounds, the same cut of 3 ng/mL was used for all men regardless of age where the proportion of positive tests ranged from 9.6% to 13.9% between screening rounds. A similar positivity rate was reported in the United Kingdom (11%) using the same PSA cut-off among 50–69-year olds [15] and positivity rates between 8% [16] and 17% [17] were reported from studies using a cut-off 4 ng/mL. Percentage of biopsied PSA test-positive persons in our cohort ranged from 28.4% to 40.0%. A low percentage of compliance to biopsy may show the use of urologist evaluation for other conditions, which may elevate PSA concentration, or the use of additional assessment techniques, e.g., multiparametric magnetic resonance imaging (MRI), free/total PSA ratio, and PSA density, which clinically allowed to decide against prostate biopsy [18]. The percentage of biopsied test-positive (>4 ng/mL.) and test-positive/abnormal DRE persons in the PLCO trial were 30.1–40.2% and 23.5–31.0%, respectively [16]. Total average biopsies/positive test ratio among ERSPC centres was 85.6% [19].

The percentage of cancer among screened persons in our cohort varied from 1.4% to 1.9%, with the average rate of 1.6%. Other countries with prostate cancer screening activities have demonstrated similar prostate cancer detection rates, e.g., in Japan, municipality-based prostate screening detection rate was 0.5–1.1% [13], while the Finland section of ERSPC achieved a 2.5% detection rate with a PSA cut-off level of 4 ng/mL during first three years of the study, whereas the detection rate in the Dutch centre was 4.6%, and in the Italian centre 1.6% of men were screened [20,21]. In the initial four rounds of screening in the PLCO trial, the detection rate for prostate cancer was 4.9% [16]. The proportion of men diagnosed with prostate cancer in the intervention group was 4.3% in England and Wales [15].

The proportion of prostate cancer among PSA test-positives that is a positive predictive value (PPV) of the screening has been reported so far in two different ways, by dividing screen detected cancers by the number of biopsies (ERSPC trial) and by the number of PSA test-positives (PLCO trial) [16,17]. PPV as a proportion of prostate cancer among biopsied individuals ranged from 35.3% to 42.0%, and ratio between screen-detected cancer and PSA-positive varied from 9.6% to 13.9%. PPV of biopsy among ERSPC study centres in different periods was 21.7–30.2% in Finland, with average ratio among centres being 24.3% [19]. PPV for test positive in PLCO trial among patients with elevated PSA and abnormal DRE findings, has been smaller and decreased from 11% at initial screening round to 7.3% in the fourth round of the screening. PPV for biopsy at the initial screening round was 36.9% and remained at the level of 31.0% in following screening rounds [16].

A positive effect with so-called “stage migration” to a localized form of disease has already been observed in our previous study [22]. Patients in the national Lithuanian prostate cancer cohort were more likely to be diagnosed with localized disease. A higher proportion of organ-confined disease was also observed in the screening group in other studies reporting stage distribution [16,23,24,25]. The proportion of localized disease is strongly influenced by overdiagnosis, which, according to the literature, is estimated to range from 1.7% to 67% [26].

Our study has several strengths and limitations to be considered. EPCDP is a population-based ongoing prostate cancer screening programme that is unique in this context and reflects real world data, like in an observational study performed in Austria [27]. Data corresponding to the screening programme were collected by a single institution covering more than 98% of country residents. However, some of the major data from the screening were not considered for the registration (e.g., exact PSA concentration, DRE results during urological assessment, and Gleason scores in biopsy were not available). Moreover, a linkage of PSA testing and follow-up procedures outside of the screening programme was not available, making estimation of the screening programme performance somewhat limited.

## 5. Conclusions

Despite the unorganized screening practices being employed and low coverage per screening round, 70% of men aged 50–74 years have participated in the prostate cancer programme at least once in the first 10 years of the screening. During the study period, in the target population, 94.8% of prostate cancer cases were detected among those who participated in the screening programme. Men participating in the screening were more likely to be diagnosed with organ-confined disease as compared to those not participating.

## Figures and Tables

**Figure 1 jcm-09-03826-f001:**
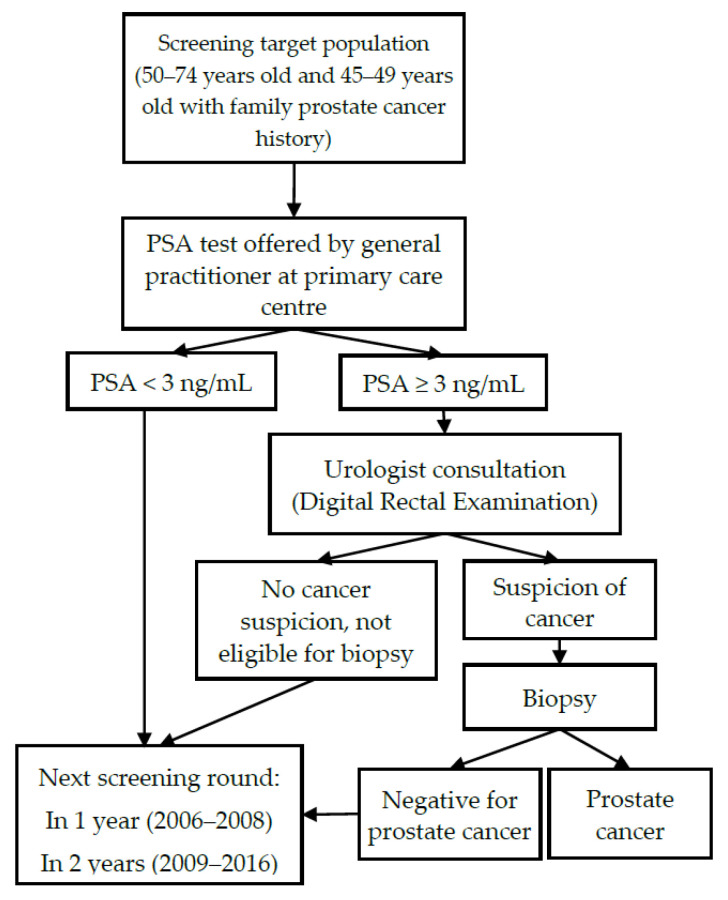
Organizational structure of Early Prostate Cancer Detection Programme in Lithuania between 2006 and 2016; PSA, prostate specific antigen.

**Figure 2 jcm-09-03826-f002:**
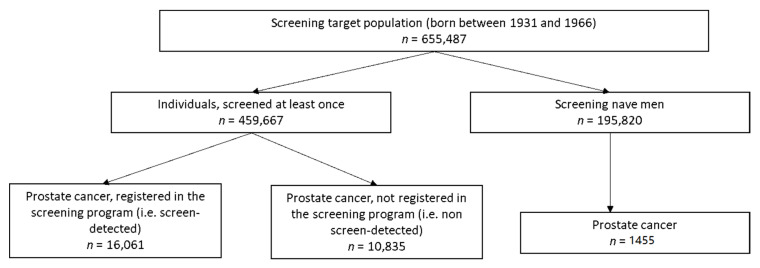
Prostate cancer diagnosis among individuals aged 50 to 74 years in Lithuanian between 2006 and 2015.

**Figure 3 jcm-09-03826-f003:**
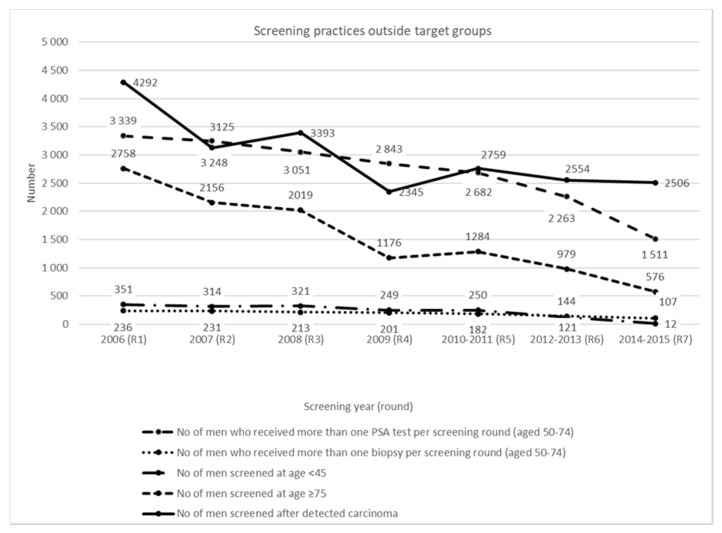
Screening practices outside the target population in the first seven screening rounds (2006–2015).

**Figure 4 jcm-09-03826-f004:**
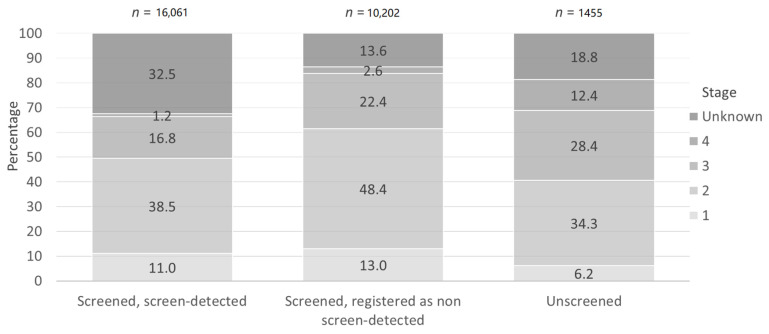
Cancer stage distribution in unscreened, screened non-screen detected and screened, screen detected groups.

**Table 1 jcm-09-03826-t001:** Main performance indicators of of Early Prostate Cancer Detection Programme (EPCDP) in Lithuania in the first seven screening rounds between 2006 and 2015.

	Calendar Year (Screening Round)						
	2006 (R1)	2007 (R2)	2008 (R3)	2009 (R4)	2010–2011 (R5)	2012–2013 (R6)	2014–2015 (R7)
Target population	413,997	417,832	422,812	429,535	466,557	480,194	492,291
Individuals screened (50–74-year-old)	92,896	99,556	121,871	97,407	184,213	200,079	223,958
Coverage (participation rate, %)	22.4	23.8	28.8	22.7	39.5	41.7	45.5
PSA results							
PSA < 3 ng/mL (%)	77,188 (83.1)	84,201 (84.6)	105,303 (86.4)	84,666 (86.9)	162,806 (88.4)	176,939 (88.4)	199,968 (89.3)
PSA ≥ 3 ng/mL (%)	15,708 (16.9)	15,355 (15.4)	16,568 (13.6)	12,741 (13.1)	21,407 (11.6)	23,140 (11.6)	23,990 (10.7)
Biopsy							
Number of biopsies (% of PSA test-positive)	4459 (28.4)	5574 (36.3)	5934 (35.8)	5092 (40.0)	8386 (39.2)	8750 (37.8)	7985 (33.3)
Prostate cancer (% of biopsy)	1509 (35.9)	1873 (36.1)	1879 (35.3)	1647 (35.9)	2836 (37.6)	3210 (40.2)	3107 (42.0)
% prostate cancer of PSA test-positive	9.6	12.2	11.3	12.9	13.2	13.9	13.0
% prostate cancer of screened persons	1.6	1.9	1.5	1.7	1.5	1.6	1.4
Prostate cancer (among screened)	2445	3320	3242	2912	4796	5143	5038
Cancer detection rate	2.5	3.2	2.69	2.9	2.5	2.5	2.2

PSA, prostate specific antigen; R, round.

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
