# Peer review of "Prostate Cancer Screening with PSA: Ten Years’ Experience of Population Based Early Prostate Cancer Detection Programme in Lithuania"

_jcm, 2020, doi:10.3390/jcm9123826_

Round 1
Reviewer 1 Report
This paper describes the results of the first 10 years of the first PSA screening programme in the world. The paper provides a complete and clear summary of the results of the programme, as far as data are available.
The writing in the English language can be improved.
Specific remarks:
- Introduction line 36: this description of the situation in the US is not totally correct. There is (and was) no population-based screening in the US. In 2011 indeed the USPSTF issued a recommendation against screening and as a result the (opportunistic) screening coverage dropped. In 2018 this recommendation was updated to shared decision making for men age 55-69. Please correct this description.
- Results: The target age range is quite broad. Are results available by 5-year age group?
- Results Figure 4: The stage of 32% of the screen detected cases is unknown. Why is this so much higher than for the other cases? The shift in stage distribution is therefore difficult to assess.
- Discussion line 158: Can you briefly describe whether there is a decline in prostate cancer mortality?
- Discussion line 196: Detection rates of other centers of the ERSPC can be found in Otto et al, Eur J Cancer 2010. Please cite more than only Finland.
Author Response
Reviewer #1:
- Introduction line 36: this description of the situation in the US is not totally correct. There is (and was) no population-based screening in the US. In 2011 indeed the USPSTF issued a recommendation against screening and as a result the (opportunistic) screening coverage dropped. In 2018 this recommendation was updated to shared decision making for men age 55-69. Please correct this description.
Answer: Authors have corrected description of prostate cancer screening in US according to reviewers comment.
- The target age range is quite broad. Are results available by 5-year age group?
Answer: We have analyzed screening activity and results by 5-years age group and have found increasing number of test-positives, biopsies and cancers detected with age. Screening activity was similar in all age groups. As this information provides no additional information and reporting these results looks redundant, we have decided to report results for target age group and 45-49 years age group (as this group could reflect screening for men, having family history of prostate cancer).
- The stage of 32% of the screen detected cases is unknown. Why is this so much higher than for the other cases? The shift in stage distribution is therefore difficult to assess.
Answer: Unknown stage of disease is higher due to completeness of Cancer registry data and reporting issues to Cancer registry. Based on Cancer survival in Lithuania analysis (Krilaviciute et al. 2014, Gondos et al. 2015), unknown stage of disease seems to be distributed evenly between stages.
(see table in attachment).
Therefore we think this figure clearly represents shift in stage distribution.
- Can you briefly describe whether there is a decline in prostate cancer mortality?
Answer: Mortality trends in Lithuania are described in another paper by our group (Patasius et al., Prostate cancer incidence and mortality in the Baltic states, Belarus, the Russian Federation and Ukraine // BMJ open. 2019, 9, 10, e031856, p. 1-7. DOI: 10.1136/bmjopen-2019-031856). Mortality in Lithuania started to decline in 2007. As mortality decline observed one year after PSA based screening programme start, obviously it is attributable for PSA test use implementation in clinical practice, but not for PSA screening.
- Detection rates of other centers of the ERSPC can be found in Otto et al, Eur J Cancer 2010. Please cite more than only Finland.
Answer: Thank you for your comment. We have cited range of detection rates, observed in ERSPC centres and added Otto et al. EurJCancer 2010 publication in the reference list.
The writing in the English language can be improved.
Answer: typos and grammar were corrected.

Reviewer 2 Report
- The objective of the screening program is not defined in the paper. Please add it.
- Benefit of screening is not diagnosis, but reduced cause-specific mortality. The statement in the discussion “reduction in mortality could be regarded as one of the success indicators” is misleading, as mortality reduction is the one and only valid measure of screening benefit. The results reported here are nothing more than process indicators (reflecting intermediate outcomes, but not the real aim of screening). This is not properly articulated in the manuscript.
- This report provides some evidence about harms but none about possible benefits (which remain uncertain). The major harm of cancer screening in general, and PSA screening for prostate cancer especially, is overdiagnosis of disease that would not have been diagnosed in the absence of screening. Please add estimates of overdiagnosis from the literature (if unable to estimate it from these data).
- Proportion of localized cases is not a valid indicator of screening benefit due to overdiagnosis. Instead, risk of advanced disease as number of cases relative to population size should be given. Please add numbers of cases and population sizes to Figure 4 and report the risk (number of cases relative to population size) in the text. A possible reduction in risk of advanced disease is the only measure of benefit that can be obtained form the present data.
- Correspondingly, some indication of overdiagnosis can be obtained by comparing the cumulative incidence of localized disease between screened and non-screened men and this should be shown in the paper. Of course the current estimate will be affected by lead-time, with catch-up in the non-screened population expected.
- There is a grave misunderstanding of the screening terminology already in the manuscript title: This is not a population-based program if it is performed on an ‘invitation by opportunity basis’ through GPs patient contacts (nor was there ever a population-based screening in the US, unlike stated in the Introduction). Here, screening is not offered systematically to everyone in the target population. This is misleading and must be corrected. This should also be indicated in the Figure 1
A related issue is the description of the program: the authors state that “PSA test was offered for all men aged 50-74 years”, which strongly contradicts the description above. If a PSA test was offered only for men who had a contact with GPs, it means that not all men were offered a PSA test. Please modify the flow chart by adding a box/level indicating the GP's role.
- Were the PSA tests centralized and standardized? Please specify
- Were the authors able to distinguish between PSA and biopsies among asymptomatic men in the program and those performed due to symptoms (LUTS)? If not this should be added as a weakness
- Please report numbers of men broken down by numbers of screens
MINOR
- In the Introduction, evidence about benefits and harms is not cited, only policy based on the evidence. Please add findings of the randomized screening trials or evidence summaries.
- As for terminology, the authors refer to men not screened within the program as ‘screening-naïve’, which is not accurate, as they may have been tested outside the program (unless they were grouped separately)
- Offering a test to men with a previous diagnosis of prostate cancer is not screening, and those men should be removed from the analysis
- I find it odd that the authors speculate about use of MRI among screen-positive men in the discussion, as this was not part of the program, and I strongly suspect that the availability of prostate MRI in Lithuania in 2006-2015 must have been minimal
- The discussion on the relatively low detection rate compared with other studies is thin. Why was the cancer rate low?
- It would be appropriate to cite earlier screening programs such as that in Austria
Author Response
Reviewer #2:
- The objective of the screening program is not defined in the paper. Please add it.
Answer: According to reviewer comment, we have defined objective of the screening program in Introduction part of this manuscript.
- Benefit of screening is not diagnosis, but reduced cause-specific mortality. The statement in the discussion “reduction in mortality could be regarded as one of the success indicators” is misleading, as mortality reduction is the one and only valid measure of screening benefit. The results reported here are nothing more than process indicators (reflecting intermediate outcomes, but not the real aim of screening). This is not properly articulated in the manuscript.
Answer: thank you for your comment. As incidence and mortality trends in Lithuania we have analyzed in our previous paper (Patasius et al., Prostate cancer incidence and mortality in the Baltic states, Belarus, the Russian Federation and Ukraine // BMJ open. 2019, 9, 10, e031856, p. 1-7. DOI: 10.1136/bmjopen-2019-031856), aim of this paper is to provide performance indicators of this programme. According to your comment, we have corrected statement about prostate cancer success indicators.
- This report provides some evidence about harms but none about possible benefits (which remain uncertain). The major harm of cancer screening in general, and PSA screening for prostate cancer especially, is overdiagnosis of disease that would not have been diagnosed in the absence of screening. Please add estimates of overdiagnosis from the literature (if unable to estimate it from these data).
Answer: Aim of this study is to report key performance estimates from the ten years of population-based prostate cancer screening programme in Lithuania, therefore we not discussed in details overdiagnosis reported in literature.
- Proportion of localized cases is not a valid indicator of screening benefit due to overdiagnosis. Instead, risk of advanced disease as number of cases relative to population size should be given. Please add numbers of cases and population sizes to Figure 4 and report the risk (number of cases relative to population size) in the text. A possible reduction in risk of advanced disease is the only measure of benefit that can be obtained form the present data.
Answer: As risk of advanced disease analysis is our future plans and out of scope of this paper, we decided to report stage distribution based on objectives of screening program.
- Correspondingly, some indication of overdiagnosis can be obtained by comparing the cumulative incidence of localized disease between screened and non-screened men and this should be shown in the paper. Of course the current estimate will be affected by lead-time, with catch-up in the non-screened population expected.
Answer: Cumulative incidence and cumulative mortality analysis among screened and non-screened patient arms is in a scope of our next paper. Aim of this study is was to report key performance estimates of population-based prostate cancer screening programme in Lithuania.
- There is a grave misunderstanding of the screening terminology already in the manuscript title: This is not a population-based program if it is performed on an ‘invitation by opportunity basis’ through GPs patient contacts (nor was there ever a population-based screening in the US, unlike stated in the Introduction). Here, screening is not offered systematically to everyone in the target population. This is misleading and must be corrected. This should also be indicated in the Figure 1.
Answer: Thank you for your comment and correction regarding prostate cancer screening in US. We have corrected statement about prostate cancer screening in US. Although, we can‘t completely agree regarding type of prostate cancer screening in Lithuania. As prostate cancer screening program is issued, administrated and promoted by national healthcare insurance fund, which covers about 98 percents of population. Ninety-five percent of all prostate cancer cases in Lithuania are diagnosed in setting of prostate cancer screening program. So, it tend to be population-based. Initially it had plans to have invitation system, although, implementation of this program became responsibility of primary care centres.
According to your comment, we have corrected Figure 1.
- A related issue is the description of the program: the authors state that “PSA test was offered for all men aged 50-74 years”, which strongly contradicts the description above. If a PSA test was offered only for men who had a contact with GPs, it means that not all men were offered a PSA test. Please modify the flow chart by adding a box/level indicating the GP's role.
Answer to this comment is covered with an answer to comment No.7.
- Were the PSA tests centralized and standardized? Please specify
Answer: PSA tests were not centralized as it was done in clinical trials. Although, all laboratories performing PSA tests analysis are standardized according to ISO-15189 standard.
- Were the authors able to distinguish between PSA and biopsies among asymptomatic men in the program and those performed due to symptoms (LUTS)? If not this should be added as a weakness
Answer: Lower urinary tract symptoms (LUTS) are more specific to benign prostatic obstruction. LUTS or bone pain could be observed only in advanced stage of disease. In this study we analysed information on PSA tests reported as tests within screening program.
- Please report numbers of men broken down by numbers of screens
Answer: This comment is not clear, therefore is not taken into account.
MINOR
- In the Introduction, evidence about benefits and harms is not cited, only policy based on the evidence. Please add findings of the randomized screening trials or evidence summaries.
Asnwer: we added short evidence summary of prostate cancer screening in Introduction part of manuscript. This issue is covered by answer to comment 3.
- As for terminology, the authors refer to men not screened within the program as ‘screening-naïve’, which is not accurate, as they may have been tested outside the program (unless they were grouped separately)
Answer: By term ‘screening-naïve’ we refer to men, who were not reported as participants in prostate cancer screening program.
- Offering a test to men with a previous diagnosis of prostate cancer is not screening, and those men should be removed from the analysis
Answer: Patients with previously diagnosed prostate cancer were removed from further analysis. As screening registry is not available in Lithuania, some proportion of prostate cancer screening services probably were used for diagnostic purposes. We have provided numbers of screens outside the target group to reflect unorganized nature of prostate cancer screening program.
- I find it odd that the authors speculate about use of MRI among screen-positive men in the discussion, as this was not part of the program, and I strongly suspect that the availability of prostate MRI in Lithuania in 2006-2015 must have been minimal.
Answer: Patients are evaluated for all other conditions, which may elevate PSA concentration. We have mentioned MRI, as an example additional asessment techniques, clinically allowed to decide not perform prostate biopsy. We have specified techniques, used to decide not perform prostate biopsy.
Statement that, MRI availability in Lithuania in 2006-2015 was minimal is not correct, as number of MRI machines, according to Statistics of Lithuania, increased from 10 to 32 and this number for 3 million population is adequate.
- The discussion on the relatively low detection rate compared with other studies is thin. Why was the cancer rate low?
Answer: As compliance to biopsy in other studies were higher, detection rates of prostate cancer were higher as well. As we have specified the range of cancer detection rates in ERSPC centres, cancer detection rates are very similar to rates, observed in Italian ERSPC centre.
- It would be appropriate to cite earlier screening programs such as that in Austria
Answer: we mentioned results of Autrian screening programme and cited in the manuscript according to reviewer suggestion.
English language and style minor spell check required
Answer: typos and grammar were corrected.
